# Readiness of public health facilities to provide quality maternal and newborn care across the state of Bihar, India: a cross-sectional study of district hospitals and primary health centres

Japneet Kaur,[1] Samuel Richard Piers Franzen,[2,3] Tom Newton-Lewis,[2] Georgina Murphy[4,5]

[1]Oxford Policy Management, Delhi, India
[2]Oxford Policy Management, Oxford, UK
[3]Centre for Tropical Medicine and Global Health, University of Oxford, Oxford, UK
[4]Health Services Unit, Kenya Medical Research Institute/ Wellcome Trust Research Programme, Nairobi, Kenya
[5]Bill and Melinda Gates Foundation, Seattle, Washington, USA

**Correspondence to**
Japneet Kaur;
japneetkaur58@gmail.com

## ABSTRACT

**Introduction** Poor access to quality healthcare is one of the most important reasons of high maternal and neonatal mortality in India, particularly in poorer states like Bihar. India has implemented initiatives to promote institutional maternal deliveries. It is important to ensure that health facilities are adequately equipped and staffed to provide quality care for mothers and newborns.

**Methods** We conducted a cross-sectional study of 190 primary health centres (PHCs) and 36 district hospitals (DHs) across all districts in Bihar to assess the readiness of facilities to provide quality maternal and neonatal care. Infrastructure, equipment and supplies and staffing were assessed using the WHO service availability and readiness assessment and Indian public health standard guidelines. Additionally, we used household survey data to assess the quality of care reported by mothers delivering at study facilities.

**Results** PHCs and DHs were found to have 61% and 67% of the mandated structural components to provide maternal and neonatal care, on average, respectively. DHs were, on average, slightly better equipped in terms of infrastructure, equipment and supplies by comparison to PHCs. DHs were found to be inadequately prepared to provide neonatal care. Lack of recommended handwashing stations and bins at both DHs and PHCs suggested low levels of hygiene. Only half of the essential drugs were available in both DHs and PHCs. While no association was revealed between structural capacity and patient-reported quality of care, adequacy of staffing was positively associated with the quality of care in DHs.

**Conclusion** Examining all DHs and a representative sample of PHCs in Bihar, this study revealed the gaps in structural components that need to be filled to provide quality care to mothers and newborns. Access to quality care is essential if progress in reducing maternal and neonatal mortality is to be achieved in this high-burden state.

## Strengths and limitations of this study

► The findings of this study are based on data collected from a single visit to these facilities; the availability of different equipment and supplies might vary over time.

► There is incomplete data in some facilities. Therefore, the number of responses varied across and within the components of infrastructure, supplies, equipment and staffing.

► With respect to household data, women providing information on quality of care were not representative of those delivering at facilities and the sampling was not proportional to the number of deliveries at each facility.

► This study is limited to assessing the structural capacity of the facilities to deliver quality care as reported by the mothers. However, there could be multiple other components that influence quality of care (eg, skills and competencies of health personnel delivering care) that were not explored in this study.

► The study covers all district hospitals and a large representative sample of primary health centres in Bihar. To our knowledge, no study of this scale has previously been conducted on facility readiness for maternal and newborn care in Bihar.

## INTRODUCTION

Progress has been made in reducing maternal and newborn mortality in India over the last three decades. Between 1990 and 2015, maternal mortality reduced from 556 to 174 per 100 000 live births and neonatal mortality reduced from 54 to 29 per 1000 live births.[1–3] However, considerable further improvements will be needed if India is to reach the Sustainable Development Goal of reducing maternal mortality to less than 70 per 100 000 births and neonatal mortality to at least as low as 12 per 1000 live births by 2030.[4]

These goals will be particularly challenging for Bihar, the third most populated state in India (approximately 104 million). Bihar

struggles with persistent poverty (34% of the population lives below the poverty line) and poor health outcomes (neonatal mortality rate of 27 per 1000 live births and maternal mortality rate of 208 per 100 000 live births).[5–7] Only 63% of the pregnant women deliver in a health facility in Bihar, which is 12% lower than the national average.[8 9] This is an important area that needs attention since the biggest gains in survival are estimated to be achieved through facility-based maternal care provided at the time of childbirth and the immediate postpartum period along with newborn care.[10]

In recognition of the importance of facility-based maternal and newborn care, India has implemented many initiatives to encourage institutional deliveries. The most ambitious of these is the Janani Suraksha Yojana (JSY) programme offering conditional cash transfers to women of low socioeconomic status for delivering at a health facility.[2] Despite the success of JSY in increasing institutional deliveries, provision of quality care has been highlighted as an important barrier for the programme to have the intended effect on health outcomes.[11 12] Addressing the gaps in facility readiness has been considered as an important factor in improving quality of care.[13] Poor availability of clinical services due to lack of infrastructure has been recognised as one of the most common bottleneck to providing essential maternal and newborn services in India.[14 15] Rammohan *et al* report lack of transport facilities for pregnant women as one of the major bottlenecks to access emergency obstetric care in India.[15] Capacity in terms of equipment and staff availability needs to be built to detect and manage obstetric emergencies.[16]

It is crucial to ensure that facilities are adequately resourced and equipped to deliver essential maternal and newborn care.[14 17–19] The role of quality factors such as infrastructure, equipment, supplies and staffing is acknowledged, but little research has been done to quantify and describe these gaps in detail. This is needed if interventions to strengthen quality are to be appropriately designed and targeted to be effective.[20]

The public health system in India comprises of a three-tier system, namely, primary care at the village level, secondary care at the sub-district and district levels and tertiary levels of healthcare at the regional level. The district hospital (DH) is an essential medium of secondary level of healthcare with an objective to provide curative, preventive and promotive healthcare services to the people in the district. Linked to every DH are health centres providing primary care, including subdivisional hospitals, community health centres (CHCs), primary health centres (PHCs) and sub-centres. PHCs are crucial to the health system as they form the first point of contact to a qualified doctor of the public sector for the patients. There are two kinds of PHCs, one is called additional PHC which mainly does clinical work and the other is block PHC which also exercises administrative powers in the entire block. Serving a population between 20 000 and 30 000, PHCs act as a referral unit for six sub-centres

and refer out cases to CHCs and higher order facilities. In Bihar, there are 36 DHs, 70 CHCs, 9729 sub-centres and 1883 PHCs (including 534 block PHCs).[21]

This study aims to (a) assess and highlight structural and staffing gaps in the public health facilities, specifically, PHCs and DHs in Bihar, that need to be addressed, to deliver quality maternal and newborn services and (b) understand the relationship between structural and process quality metrics for maternal and newborn health services. This study is based on the data collected in the baseline assessment of Bihar Technical Support Programme (BTSP). BTSP is a large multi-year programme funded by the Bill and Melinda Gates Foundation and implemented by CARE India with Oxford Policy Management (OPM) as monitoring and evaluation partner.[22] Working closely with Government of Bihar's Departments of Health and Family Welfare and Social Welfare, CARE India's interventions aim to strengthen the health system and improve the quality of care to improve reproductive, maternal, newborn, child, adolescent and nutrition (RMNCH+N) outcomes.

## METHODS
We conducted a cross-sectional study of health facilities in Bihar during July 2016 to October 2016. Facility surveys were conducted in block (subdistrict) and district level government-run public health facilities. This study also uses household maternal and child health survey data collected during October 2016 to December 2016 by CARE India.

### Study population and sampling
There are 36 district hospitals in Bihar, all of which were invited to participate in the facility survey. There are 534 blocks (subdistricts) in Bihar, 190 of which were sampled for the facility survey. The number of blocks vary widely per district. Hence, blocks were sampled proportionally according to the total number of blocks per district. The selected sample had blocks ranging from one to nine per district with a median of six blocks. Each block contains one block PHC, all of which (from the 190 sampled blocks) were included in the facility survey.

Household survey data were collected using five different questionnaires for mothers who had a child belonging to the following five age groups: (i) 0 to 2, (ii) 3 to 5, (iii) 6 to 8, (iv) 9 to 11 and (v) 12 to 23 months old. A mixed sampling methodology of population based-estimation and lot quality assurance sampling (LQAS) (a small sample survey design based on binomial distribution) was used.[23] The sampling 'lots' in this survey were the blocks/subdistricts. All 534 blocks in 38 districts were included in the study data collection. The number of anganwadi centres (AWCs, village level institutions providing basic healthcare services) sampled from each block was determined using proportional allocation, however if this resulted in a sample of less than 19 AWCs, then 19 AWCs were sampled in order to meet a

minimum sample threshold per block. The sampled AWC were selected within each block using simple random sampling. Five households per AWC were selected, with one each from mother of following five age groups: (a) 0 to 2, (b) 3 to 5, (c) 6 to 8, (d) 9 to 11 and (e) 12 to 23. In total, 15 667 AWCs were selected ranging from 19 to 123 per block.

Within each sampled AWC catchment area, households were identified through systematic sampling.[23] Briefly, an index household was chosen within each AWC catchment area using a random number table. Starting with the index household, data collectors visited every fifth household looking for eligible mothers. This approach aimed to obtain a wide distribution of households (minimising the effect of clustering), while remaining feasible and practical for data collection purposes. The pilot phase of the study did not observe any significant differences in household characteristics when alternative sample intervals of 10th, 15th and 20th households were selected. The data collectors continued moving in a circular manner, following the 'right-hand rule', until five eligible households had been interviewed per AWC catchment area, one household for each age group questionnaire.

To reduce the recall bias, data on quality care presented in the analysis were restricted to mothers with children aged between 0 to 2 months. Of the mothers who also delivered at the DHs or PHCs that were covered in the facility survey (ranging from 1 to 17 mother per facility) were included in this analysis.

## Data collection

### Facility survey

Data were collected using a standardised structured survey tool designed based on the Service Availability and Readiness Assessment tool developed by the WHO and the United States Agency for International Development.[24] The tool was modified for the Indian context using the Indian Public Health Standards (IPHS) guidelines.[25 26] To evaluate the structural capacity of the facility, the availability and condition of infrastructure, equipment and supplies in different departments, including the labour room, newborn care corner , immunisation room, laboratory, operation theatre, drug store and data operation were assessed. Information on infrastructure and equipment was collected through interviews with the facility-in-charge and staff nurse as well as through direct observation. The pharmacist or drug store-in-charge was interviewed, and the responses were validated through the drug register to collect information on supplies availability.

The medical officer in charge (MOIC) at the PHCs and hospital manager at the DHs were also interviewed to obtain information on the number of health personnel employed at the facilities and the number of personnel that were sanctioned (number of staff expected to be employed) to the facilities for each of the health cadres, including medical officers (MOs), staff nurses, auxiliary nurse midwife (ANM), laboratory technicians and pharmacists. This information was also cross-checked with the facility registers.

Availability of 30 services related to family planning, safe delivery, antenatal care and neonatal and child care was assessed and the reasons for unavailability were asked from the MOIC in PHCs and the hospital manager in DHs.

Three pilot tests were conducted in the facilities outside the study sample to refine the survey tool and to train the enumeration team. The survey was conducted by 60 enumerators over the 4 month period. Enumerators all had prior experience in conducting facility surveys and received further training over 10 days on using the study tool and conducting this survey.

Periodic data checks for completeness and outliers were conducted by a data management team in Patna, Bihar. Where information was missing due to absenteeism or lack of time provided by the respondent, a second visit to those facilities was organised.

### LQAS household survey

One-to-one interviews were conducted with consenting and eligible mothers by trained data collectors, using a standardised questionnaire and following standard operating procedures. Information collected from mothers and of interest to this study included the household characteristics, the place of delivery and care received at the place of delivery.

### Patient and public involvement

Patients were not involved in the study.

### Data analysis

Data analysis was conducted using Stata V.13 (Stata Corporation, USA). The current status of the facilities was assessed on three broad parameters, namely, the structural capacity, staffing and the quality of care provided at the facilities.

### Structural capacity

The structural capacity of the facilities was assessed by computing readiness scores of 0 to 1 for infrastructure, equipment and supplies. 'Infrastructure readiness' included the availability as well as the condition of different components, wherever applicable. For equipment, 'readiness' implied the availability as well as functionality of the equipment and for supplies, readiness was defined by availability.[24]

Infrastructure readiness of the facilities included nine broad components (such as power, water, transport, handwashing stations) at the PHCs.[24] An additional three components (availability of different rooms, computer and internet and blood bank) were assessed for DH infrastructure score (details of components are listed in online supplementary table S1).

The equipment readiness of the facilities was assessed by scoring the availability and functionality of 48 essential (according to IPHS guidelines) maternal and newborn health equipment (items listed in online supplementary

table S2). A score of 1 was assigned if the equipment was observed to be available and in a functional state. In case of unavailability or available but not functional equipment, a score of 0 was assigned. Similarly, supplies readiness was assessed by considering the availability of 76 essential maternal and child health drugs that were expected at the facilities as per the IPHS guidelines and contextualised based on the requirements in Bihar (listed in online supplementary table S3). The mean across the three components of infrastructure, equipment and supplies was computed to generate a score for structural capacity ranging from 0 to 1 per facility. The mean across facilities was computed to get an overall score for structural capacity. Detailed methods of scoring have been provided in the online supplementary data.

## Staffing index

We assessed the availability of human resources by computing the ratio of filled to sanctioned positions, as reported by the MOIC and the hospital manager or equivalent authority in charge in the PHCs and DHs, for each health cadre in each facility. The ratio of total filled to total sanctioned positions for permanent staff, combining all cadres, was computed to generate an overall staffing index for each facility.

The availability of health staff was also compared with the essential requirements mandated by IPHS guidelines. In PHCs, we considered staff requirement based on the monthly delivery load of more than 20, as provided by the IPHS guidelines.[25] In DHs, the staff requirement based on the bed strength were rounded down to compare with the mandated guidelines.[26] For instance, for DHs with less than or equal to 200 beds, we considered the staff requirements for 100 beds as defined by IPHS guidelines. For ANMs, the IPHS requirement of 0.45 staff per bed was considered. (online supplementary table S4).

The relationship between availability of services (that were unavailable in at least 10% of the PHCs and DHs) and structural capacity and staffing index was explored by assessing the pairwise correlation coefficients between the indices at the facility.

## Quality of care

Our primary aim was to describe the structural readiness of facilities to provide essential maternal and newborn services. We also conducted analyses of household survey data to explore the quality of care at facilities as reported by women who both participated in the household survey and delivered at study facilities.

Each mother was asked 11 questions during the household survey pertaining to the treatment and care that they and their newborns received during delivery. Each question was assigned a score of 0 (not performed/ don't know) or 1 (performed). Household survey data was merged with facility data by matching the names of facilities where mothers delivered with the facility names collected during facility assessment survey. A quality of care index for each PHC and DH was generated by taking the average score of the 11 questions for all those household survey participants who delivered within the facility. All data were assessed at the facility level.

The relationship between structural capacity, staffing and quality of care indexes were visually explored using scatter plots and trend lines as part of this exploratory analysis.

## Ethics and permission

Ethical approval was granted by the Indian Institutional Review Board. At each facility, the purpose of the study was explained and informed consent was obtained from the MOIC and the hospital manager or equivalent authority in charge in the PHCs and DHs, respectively. For the household survey, ethics approval was obtained from Ashirwad Ethics Committee, Ashirwad Hospital and Research Centre, Ulhasnagar, India, and informed consent was taken from the mothers.

## RESULTS

The number of facilities assessed for each component of structural capacity and staff availability varied (range: 35 to 36 DHs and 166 to 190 PHCs) due to missing data and depending on the availability of respondents during the time of the survey (online supplementary table S5). Household survey data were available from 671 mothers who delivered in 107 of the 190 study PHCs and 1419 mothers who delivered in across all 36 study DHs.

## Facility characteristics

Most PHCs (95%) were functional for 24 hours per day, but 40% of them were not accessible throughout the year. A dedicated labour room, maternity ward, operation theatre and store room was found to be available in most PHCs (94%, 96%, 89% and 96%, respectively); an immunisation room was available in only 76% of the PHCs. While the IPHS guidelines recommend each PHC to have six beds, the number of sanctioned and available beds, as reported by the MOIC, varied. Eight PHCs reported having no beds, four of which nonetheless conducted maternal deliveries.

All DHs were found to have a dedicated labour room and maternity ward, but specialised units for antenatal care and for postnatal care were available in only 69% and 56% of the DHs, respectively. As per the IPHS guidelines, every DH should have a provision for special newborn care units (SNCUs); however, this unit was found in only 21 of the 36 DHs (58%). In DHs, the number of beds recommended by IPHS guidelines varies between 75 to 500 depending on the size, terrain and population of the district; however, in Bihar, we identified four DHs with fewer than 75 beds available.

## Availability of services

Of the 30 services assessed in 36 DHs and 189 PHCs, seven (23%) and 12 (40%) services were unavailable in at least 10% of the facilities, respectively. Most of the

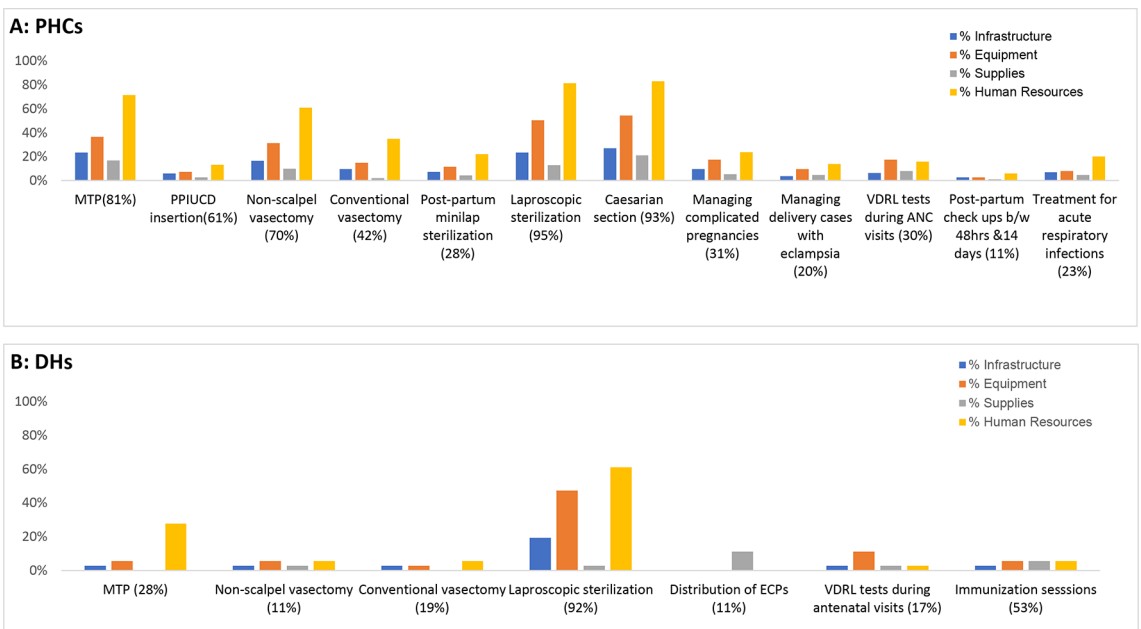

**Figure 1** Reasons for unavailability of services in (A) primary health centres (PHCs) and (B) district hospitals (DHs). Multiple answers were allowed. Figures in parentheses of x axis report the total unavailability. ANC, antenatal care; ECP, emergency contraceptive pill; MTP, medical termination of pregnancy; PPIUCD, postpartum intrauterine contraceptive device; VDRL, venereal disease research laboratory.

commonly unavailable services were related to family planning including medical termination of pregnancy (MTP), non-scalpel vasectomy, conventional vasectomy and laparoscopic sterilisation. Venereal disease research laboratory tests conducted during antenatal care visits were unavailable in 17% and 30% of the DHs and PHCs, respectively (figure 1).

For both PHCs and DHs, the main reason for the lack of these services was reported to be lack of required human resources (figure 1 and online supplementary table S6). In PHCs, lack of equipment was reported to be the second most important factor for the unavailability of services such as MTP, non-scalpel vasectomy and laparoscopic sterilisation. Lack of equipment was also the

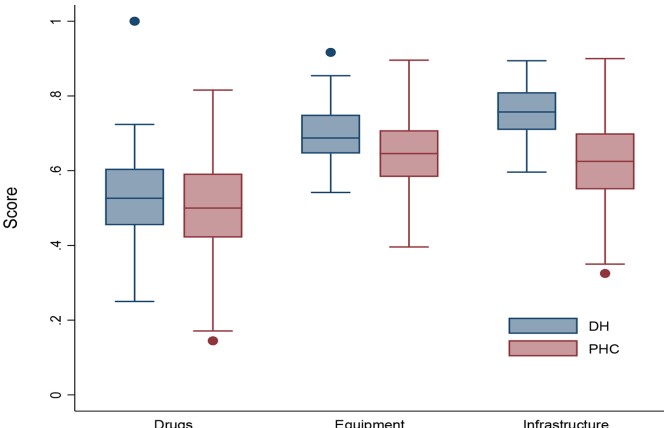

**Figure 2** Structural readiness scores across district hospitals (DHs) and primary health centres (PHCs). Scores are presented as box plots representing the median and IQR (box and whiskers, respectively) and outliers (dots).

reason for unavailability of laparoscopic sterilisation in 47% of the 36 DHs.

## Structural capacity

Overall, the average structural capacity across PHCs was 60% (range: 35% to 83%) and 66% (range: 51% to 82%) across DHs. DHs were slightly better equipped on average in terms of availability of infrastructure, equipment and drug supplies (78%, 70% and 53%, respectively) compared with PHCs (63%, 65% and 50%, respectively). Results varied greatly between facilities, particularly among PHCs (figure 2).

### Infrastructure

Infrastructure score at the DHs and PHCs varied with a range of 60% to 92% and 32% to 90%, respectively. Of the 12 items assessed in DHs, five (handwashing station in the labour room, telephone connection, water, power and transport) had an average readiness score of greater than 90%. In PHCs, telephone connection was the only component with an average readiness score of over 90% across facilities. Readiness was particularly low with respect to handwashing stations in the immunisation room and laboratory in both DHs and PHCs (online supplementary table S7).

Of all the items assessed in the labour room, the availability of different colour coded bins to segregate waste into infectious and non-infectious sources was the lowest in both PHCs and DHs (54% and 63%, respectively). Emergency transport for referrals was available in only 66% of the PHCs, whereas the DHs performed well in this regard with all DHs having emergency transport available for referrals.

## Equipment and supplies

Equipment score at the DHs ranged between 54% and 92%; the corresponding range at PHCs was 40% to 90%. Neonatal stethoscope and MTP suction were the two most commonly missing items of equipment in PHCs; whereas, in DHs, infantometer and nebuliser were the two most commonly missing items (online supplementary table S7). In the labour room specifically, light examination, feeding tube and oxygen cylinder were the most commonly missing items of equipment in both the DHs and PHCs.

Availability of drugs was the weakest performing area of structural assessment for both DHs and PHCs (figure 1), with only half (50% and 52% on average, respectively) of the essential drugs (n=76) being available. Drug score varied with a range of 25% to 100% across DHs and 14% to 82% across PHCs. Of 25 essential drugs that should be available in the labour room as per the IPHS guidelines, 62% and 72% were available on average in PHCs and DHs, respectively. Carboprost, hydralazine and methyldopa were the least commonly available of the drugs in both the PHCs and DHs.

## Staff availability

The overall average staffing index was 69% (range: 11% to 100%) in PHCs, indicating 31% of health worker sanctioned positions, as reported by the MOIC, being unfilled. The average staffing index at PHCs was found to be the highest for the ANMs, indicating a high proportion of sanctioned positions being filled (table 1). However, the requirement of ANMs, as mandated by the IPHS guidelines, was fulfilled in only 42% of the PHCs (table 1). The mandate of having at least one MO at a PHC was fulfilled at all PHCs. However, the sanctioned positions, as reported by the MOIC, varied and the average staffing index of available to sanctioned MOs was 70% for contractual (n=129) and 68% for permanent staff. The proportion of positions filled in PHCs was the lowest for laboratory technicians (27%). These technicians were, however, supplemented by contractual workers, for whom 92% of sanctioned positions were filled. RMNCH+counsellors were available in only four PHCs (2%) and family planning counsellors in six PHCs (3%). None of the PHCs had an infant and young child feeding counsellor.

In DHs, the overall staffing index for three cadres was 55% (range 24% to 100%). The staffing index among the health personnel in DHs was found to be similar to PHCs (table 1); the staffing index was also highest for ANMs (78%) and lowest (35%) for laboratory technicians in DHs. For ANMs, the IPHS requirement of 0.45 staff per bed was fulfilled in only 15% of the DHs (table 1). The average staffing index for MOs was 52% and the requirement of essential MOs as per the IPHS guidelines was fulfilled in 53% of the DHs. Nearly 60% of the DHs had less than half of the sanctioned positions for MOs and nurses filled.

## Relationship between service availability indexes

In PHCs, all three components of structural capacity index, including infrastructure, equipment and supplies, had significantly positive correlation with the availability of the 12 services at 5% level of significance. For DHs, availability of seven services that were unavailable in at least 10% DHs, had positive correlation with equipment, supplies and staffing index at 5% level of significance.

## Reported quality of care

When asked if 11 essential prepartum and postpartum services had been carried out, the responses were similar between DHs and PHCs (table 2). Almost all mothers reported that hygiene and newborn warmth practices of wearing gloves, wiping the baby dry and wrapping the baby were being practised in both PHCs and DHs. Provision of skin-to-skin contact was reported by fewer than half of women, regardless of facility type. Measuring blood pressure and advising mothers about their and their baby's health before discharge were received by less than 30% of the mothers.

**Table 1** Average filled/sanctioned positions for staff and IPHS requirement fulfilment for DHs and PHCs

| Designation | Average filled to sanctioned- DHs | Average filled to sanctioned- PHCs | % DHs fulfilling IPHS requirements | % PHCs fulfilling IPHS requirements |
|---|---|---|---|---|
| Medical officer | 52% (34) | 68% (190) | 53% (34) | 100% (190) |
| Staff nurse | 44% (33) | 42% (48) | 15% (33)* | – |
| Auxiliary nurse midwife | 78% (24) | 81% (173) | | 42% (173) |
| Laboratory technician | 35% (32) | 27% (148) | 0% (32) | 27% (148) |
| Compounder/pharmacist | 56% (32) | 63% (171) | 16% (32) | 70% (171) |
| Storekeeper | 58% (28) | 57% (101) | 61% (28) | 57% (101) |

Only permanent positions are considered. Cases where information on sanctioned positions was missing were excluded. Medical officers include physicians, obstetricians, paediatricians and anaesthetists. PHC IPHS guidelines mention to appoint at least four nurse- midwives. We consider at least four ANMs for each facility since the information for staff nurse is unavailable for most facilities.
*DH IPHS guidelines mention the requirement for staff nurse/ANM combined and hence we consider the combined availability of staff nurse and ANM.
ANM, auxiliary nurse midwife; DHs, district hospitals; IPHS, Indian public health standard; PHCs, primary health centres.

**Table 2** Quality of care reported by mothers delivering at the primary health centres (PHCs) and district hospitals (DHs)

| Quality of care | PHCs (n=671) | | | DHs (n=1419) | | |
|---|---|---|---|---|---|---|
| | Yes | No | Don't know | Yes | No | Don't know |
| Was the baby wrapped in a clean cloth after birth? | 97.91% | 1.34% | 0.75% | 96.41% | 1.20% | 2.40% |
| Did this person wear gloves before conducting your delivery? | 96.87% | 1.64% | 1.49% | 95.49% | 1.20% | 3.31% |
| Was the baby wiped dry after delivery? | 95.68% | 2.53% | 1.79% | 93.31% | 2.47% | 4.23% |
| Was the baby weighed after delivery? | 92.55% | 3.73% | 3.73% | 88.94% | 5.64% | 5.43% |
| After delivery, was nothing applied to the cord? | 91.36% | 8.67% | 0% | 85.27% | 14.73% | 0% |
| Did the person wash hands with soap before conducting your delivery? | 76.15% | 3.73% | 20.12% | 73.50% | 5.14% | 21.35% |
| Was the baby placed on the mother's abdomen immediately after birth? | 49.78% | 42.92% | 7.30% | 40.03% | 48.98% | 10.99% |
| Were you advised by the nurse or anyone else to keep the baby naked on your chest, next to your skin? | 35.77% | 63.49% | 0.75% | 23.82% | 75.26% | 0.92% |
| Did you breastfeed your baby immediately after delivery? | 24.29% | 75.71% | 0% | 21.17% | 78.48% | 0.24% |
| Was any advice given to you regarding your health or your baby's health before you were discharged from the facility? | 29.06% | 70.94% | 0% | 18.60% | 81.40% | 0% |
| Was blood pressure measured after delivery, before discharge? | 9.99% | 90.91% | 0% | 8.67% | 91.33% | 0% |

Wrapping the baby in a clean cloth after birth, wearing gloves before delivery and wiping the baby dry after delivery were the three most commonly followed practices reported by the mothers (highlighted in green). Breastfeeding the baby immediately after delivery, advice regarding mother and child's health and measuring blood pressure were the three least followed practices (highlighted in orange).

### Relationship between quality indexes

No clear relationship between the facility structural capacity index (composite score for infrastructure, equipment and drugs), the staffing index (ratio of sanctioned to filled positions) or the quality of care index (average score for 11 facility-based care services among women per facility) was found for PHCs. In DHs, no clear trend was observed between the structural capacity index and quality of care as well as staffing and structural capacity index. However, a positive relationship between the quality of care index and staffing index was evident (figure 3).

### DISCUSSION

This study provides evidence from all DHs and a large representative sample of block PHCs in Bihar, describing the gaps that need to be addressed to improve the provision of facility-based maternal and newborn care. Gaps in the structural capacity of facilities to provide quality care in terms of basic infrastructure, availability of equipment and supplies and appropriate staffing were identified. These are areas that will require coordinated and dedicated efforts if much needed gains are to be made towards improved quality of facility-based maternal and neonatal care.

The results revealed that DHs, on average, were better in terms of staffing and structural capacity in comparison with PHCs. However, the reported quality of care was better in PHCs than the DHs. DHs, being the referral points for PHCs, often need to address complicated cases and are therefore recommended to have higher staffing and structural capacity in comparison to PHCs. However, the quality of care provided at DHs and PHCs would also depend on other factors including the case load and type of cases.

The trends within the structural capacity were very similar in both the district and block facilities with availability of supplies being the lowest among the components of structural capacity. It is particularly concerning that DHs are missing drugs to control blood pressure and treat haemorrhage since they are supposed to deal with women who are at risk of complications.

Maintenance of hygiene is extremely important in clinical areas such as labour rooms with patients at high risk of acquiring infections. However, assessment of infrastructure readiness revealed a low level of hygiene and sanitation practices in the facilities. The study identified lack of recommended handwashing stations in different rooms and colour coded bins in the labour room. The establishment of a system of accreditation and regular monitoring of quality of hygienic care, among other interventions, may help to ensure that the facilities have the essential equipment and infrastructure in place.

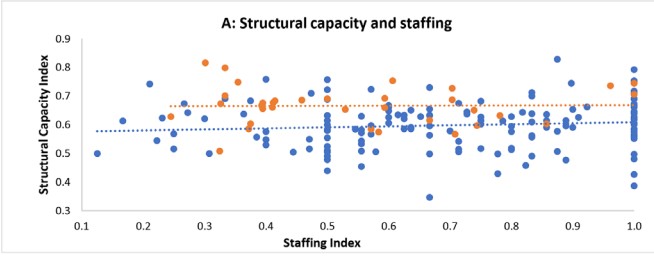

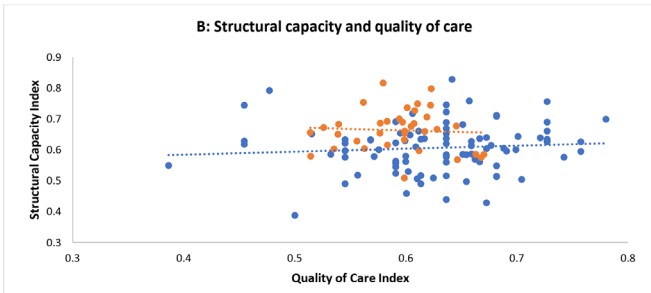

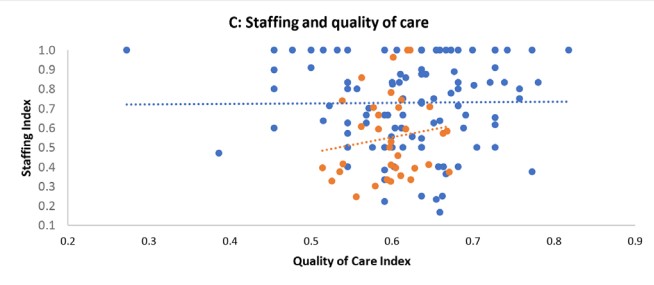

**Figure 3** Structural capacity, staffing and quality of care relationship for primary health centres (PHC) (blue) and district hospitals (DH) (orange). Each point represents the indexes for each facility (PHC or DH). The trend line shows the relationship between the staffing and quality of care across the PHCs (blue) and DHs (orange). Note that axis scales vary.

The most commonly missing equipment in the DHs and PHCs were mostly pertaining to neonatal care. Access to basic neonatal care is essential to reduce neonatal mortality because between a quarter and half of all neonatal deaths happen within 24 hours of life and 75% of neonatal deaths arise in the first week of life.[27] Preterm birth, severe infections and asphyxia have been globally identified as the main direct causes of neonatal death. Low birth weight has also been recognised as an important cause of death.[27] Low-cost interventions including tetanus toxoid vaccination, exclusive breastfeeding, kangaroo mother care for low birth weight infants and antibiotics for neonatal infections can reduce mortality.[28] However, our study revealed that skin-to-skin care was only being practised by 36% of the mothers in PHCs and 24% of the mothers in DHs across Bihar. Immediate breastfeeding practice was also reported by only 24% of the mothers in PHCs and 21% mothers DHs. These findings suggest that these facilities are not ready to provide quality neonatal care and are missing simple but vital lifesaving interventions.

Availability of skilled human resources is another important aspect to provide quality maternal and newborn care. The data on sanctioned posts, as reported by the facility in charge, were found to be different from those recommended by IPHS guidelines. This gap between the guidelines and actual sanctioned posts reflects the lack of translation of policies into practice. In PHCs, while the essential requirement for MOs was fulfilled in all facilities, the filled to sanctioned ratio was only 68%, indicating the need for more MOs in these facilities. In DHs, the IPHS requirement of staff nurse/ANM was fulfilled in only 15% of the facilities. In addition, lack of staff was reported as the main reason for the unavailability of services. Our results specifically indicated a lack of counsellors at both the block and district facilities. This may have contributed to less than 30% mothers reporting having received advice about their and their baby's health before discharge. The positive relationship found between the staffing and quality of care (as reported by mothers) at the DHs affirms the need to address the gaps in staffing to provide better quality of care.

Both DHs and PHCs are particularly important platforms under the health system, with DHs being the secondary referral level responsible for providing basic speciality services and PHCs being the first point of contact to a qualified doctor in the public health sector in rural areas. Given that the PHCs are not equipped to manage complicated cases, including caesarean sections or provide facilities of SNCU, it is important to have a well-functioning transport system for referrals. Our findings revealed that 34% of the PHCs did not have an emergency transport for referrals. While the Government of India recommends the provision of referral system at the facilities, no systematic step has been taken in this direction so far.[26 29] Lack of skilled staff, inadequate infrastructure and lack of accountability have been recognised as some of the key reasons for the failure of referral systems in India.[30]

This study has both strengths and limitations. The study draws on data from a large number of facilities, covering all DHs and a large representative sample of PHCs in Bihar. To our knowledge, no study of this scale has previously been conducted on facility readiness for maternal and newborn care in Bihar. The findings of this study are, however, based on data collected from a single visit to these facilities; the availability of different equipment and supplies might vary over time. The number of responses varied across and within the components of infrastructure, supplies, equipment and staffing, leading to incomplete data in some facilities. With respect to household data, women providing information on quality of care were not representative of those delivering at facilities and the sampling was not proportional to the number of deliveries at each facility. Hence, findings on quality of care at facilities as reported in the household survey should be treated as exploratory findings only. The scope of this study is limited to assessing the structural capacity of the facilities to deliver quality care and the care as reported by the mothers. However, there could be multiple other components that influence quality of care (eg, skills

and competencies of health personnel delivering care) that were not explored in this study.

## CONCLUSION

Presence of well-functioning facilities, with required structural and staffing capacity, is crucial for providing maternal and newborn care that translates to better maternal and child outcome. Being a highly populated state with poor health outcomes, the state of Bihar requires particular attention if India is to achieve the sustainable development goals for maternal and newborn health. This study provides a description of the current capacity of public facilities in Bihar to provide quality maternal and neonatal care, unearthing particular gaps in neonatal equipment, infrastructure required to maintain hygiene and staffing capacity at the facilities. Lack of correlation between structural capacity and staffing, and structural capacity and quality of care suggests presence of heterogeneity in the strengths and weaknesses across the facilities. A better understanding is needed to assess the cause of this variation which could help design tailored and appropriate interventions at these facilities to improve quality of care. This study lays the foundation for ongoing studies in Bihar to explore the relationship between quality of care and health outcomes. Increased focus on effective coverage and quality of facility-based care for mothers and newborns is needed if necessary gains are going to be made in saving lives in this high-burden setting.

**Acknowledgements** The authors would like to acknowledge Priya Nanda (BMGF, Delhi), Yamini Atmavilas (BMGF, Delhi), Tanmay Mahapatra and and his colleagues in the Concurrent Monitoring and Learning team of CARE India. The authors thank Sneha Lamba, Rakesh Chandra, Madhavi Misra and the entire field team of OPM for their contribution to this manuscript.

**Contributors** TN-L and SRPF conceived the idea for the manuscript. JK managed the data collection and supervised the data cleaning. JK led the data analysis and wrote the manuscript with the support of GM. JK and GM jointly interpreted the data. GM supervised and edited the manuscript. Both TNL and SF reviewed the paper. All authors were involved in the interpretation of data, critically reviewed the manuscript and approved the final version.

**Funding** This study was supported by Bill and Melinda Gates Foundation.

**Competing interests** None declared.

**Patient consent for publication** Not required.

**Provenance and peer review** Not commissioned; externally peer reviewed.

**Data sharing statement** Data are available upon reasonable request.

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
