## [Reviewer comments · BMJ Open]

ARTICLE DETAILS

TITLE (PROVISIONAL)	Readiness of public health facilities to provide quality maternal and newborn care across the state of Bihar, India: a cross-sectional study of district hospitals and primary health centres
AUTHORS	Kaur, Japneet; Franzen, Samuel; Newton-Lewis, Tom; Murphy, Georgina

VERSION 1 – REVIEW

REVIEWER	Deogratus Bintabara (MD) Department of Public Health, College of Health Sciences, The University of Dodoma, Tanzania
REVIEW RETURNED	07-Feb-2019

GENERAL COMMENTS	Thank you for the opportunity to review this informative paper. This is an important issue, and I congratulate the authors for bringing further attention to it. However, I have some slight amendments authors may wish to consider to improve the quality of the paper. General comments Despite the fact that authors made their effort in their good work, the manuscript has got some weaknesses which need extensive revision. In Introduction, authors should try to state what the research gap(s) is. In Methods, the section missing a lot of important information that could help readers to follow the manuscript. Also, the sampling procedure is not clear and missing clear information about how the outcome variable has been derived. Missing of clear research questions lead authors to presents a lot of unnecessary findings in the results section. Furthermore, there were a lot of redundancies in methods, results, and discussion sections. Finally, the conclusion is weak. If the authors revised these areas probably they would have a very nice article. Below are my comments --Abstract: 1. Conclusion: Line 4-5, "....., this study revealed that improvement in service readiness is essential...." About this sentence, I think the study has to reveal the weakness or strength of what the authors assessed regarding their research questions and not what the others think needs to be improved. 2. The same section, the second sentence "Access to quality care" despite the sentence looks ectopic but also difficult to understand what authors want to communicate. For example, reducing mortality of whom? I suggest the authors conclude based on their findings and provide recommendations on what has to be
--

	done based on their findings rather than providing a general sentence like this. --Introduction:  1. The first paragraph, Line 3, the authors forgot to report MMR and MMR appropriately. The words "live births" are missing. 2. Many of the sentences in this section do not have the supporting references, therefore it is difficult to understand whether it is the known facts or just authors arguments. 3. Paragraph 3, the authors stated that "...gaps in readiness of facilities to provide quality care have been highlighted as an important barrier...." If this is known already what was the rationale of the current study? This can be achieved by clearly stating what is known so far about readiness to provide maternal and newborn care and what is not known that your research is going to cover. --Methods: Study population and sampling  1. Page 3, second paragraph; the sentence "Household survey data were collected from" Did you use the different questionnaires for the different age groups you identified? If no, why is it necessary to indicate age groups here? If you say "less than 24 months" will it show different meaning from what you have explained by including age group categories? 2. Page 3, Line 36-7, if the blocks were sampled proportionally according to the total number of blocks per district, how did it come resulting in 5-6 blocks sampled per district? Because this seems like an equal number of blocks per each district. The sentence is confusing. 3. Page 3, Line 5-7, in the previous paragraph you stated that the blocks were sampled proportionally according to the total number of blocks per district, here again, you stated that the number of AWC from each block was determined using proportional assumption. So how many AWC did you select per each block? Did you select an equal number of AWC per each block? In total how many AWC was selected? 4. The second paragraph in the same section, data collectors visited every 10th household looking for eligible mothers. Why after every 10th household? Why not 5th, 8th or 15th? It is important to explain the justification for doing that. 5. Similarly, why did you decide to include only those women with children between 0 – 2 months? Data analysis  1. Generally, this section is not appropriately written. Instead of writing how they analyzed their data, authors just explaining the measurement of variables. 2. The section is poorly written and make it very difficult to follow. 3. In the second paragraph, the authors reported that readiness was defined as availability. However, readiness and availability are two separate components used in assessing the capacity of the facility to provide services. The authors need to provide evidence of defining readiness based on availability. 4. Also, there is a lot of redundancies, for example, the last sentence of the first paragraph and the first sentence of the second paragraph explain the same thing. 5. With the current information, it is not clear what is the outcome variable of this study. To me, it seems the outcome variable is "readiness of the facility to provide quality maternal and newborn care." However, there is no information in this paper that explains
--	--

	the overall readiness of each of the three predetermined indices or the global score. This is what I mean, based on those three indices how did you consider the facility is ready to provide the maternal and newborns care? This should be explained clearly. 6. The authors are required to provide information on how they merged the two data sets “facility dataset” and “household dataset.” Also, to state what was the unit of analysis in this study? 7. Based on your sampling technique, it seems you did a multistage (multilevel) sampling. During analysis did you consider to correct for “standard errors” as results of this kind of sampling you employed? Ethics and permission 1. The information regarding “informed consent and how was it performed” is missing. --Results: 1. Generally, missing clear research questions lead authors to presents a lot of unnecessary findings in the results section. 2. Page 6, Paragraph starting with "Facility survey data were" Please try to avoid redundancies some of the information here have also provided in the Methods section. 3. In the sections of Facility characteristics and availability of services, there is a lot of unnecessary findings presented in the text. 4. Page 9, the structural capacity has three components infrastructure, equipment, and drugs, but here you provided the average structural capacity in which you did not explain how you calculated it neither in the main document nor in the appendix. 5. Page 9 Paragraph 2 and 3, and Page 10, Paragraph 1, 2, and 3, there is a lot of repetition of information provided in the methods section. Furthermore, redundancies of results of table 1 and texts. 6. What is the relevance of assessing the relationship between structural capacity and staffing, structural capacity and quality of care, and staffing and quality of care? For example, there was a clear relationship, then, what did you want to tell the readers. --Discussion: 1. Generally, the discussion is weak and the authors failed to discuss in detail the important findings from this study instead they repeated to explain them as what they did in the results section. --Conclusion: 1. The conclusion needs to be revised to address the gap or the strength observed instead of recommending by using the general information not based on the results they obtained.
--	--

REVIEWER	Santosh Kumar Sam Houston State University, Texas, USA.
REVIEW RETURNED	20-Feb-2019

GENERAL COMMENTS	BMJ Open Manuscript ID: bmjopen-2018-028370 Summary: The authors report a descriptive analysis of readiness of primary health centers (PHC) and district hospitals (DH) to provide maternal and newborn care in Bihar, the third most populous state with poor health outcomes in India. Based on facility and household survey, the authors construct structural capacity, staffing index, and quality of care index. Surprisingly, the authors did not find any clear correlation between facility structural capacity index, staffing index, and quality of care.
---

	The manuscript is well written and reports the survey findings on readiness of health clinics that are essential for improving maternal and newborn care. Comments:  1. Introduction needs to be strengthened by including some empirical evidence on the facility readiness and facility volume of services. Since the results show correlation between structural capacity & staffing / quality of care etc, authors should report relevant literature on these associations. As of now, it lacks literature on these correlations. 2. How the average structural capacity is across PHC and DH constructed? Is it the simple average of the three scores, namely, drugs, equipment, infrastructure. Could it be greater than one? What is the range of the structural capacity index? Please clarify this on page 9, line 5-6. 3. The facility survey data are from 2016. A similar facility survey was done in District Level Household Survey (round 3) in 2007-08. It would be interesting to see the trend in facility readiness over time. A comparative analysis of DLHS 3 and BTSP survey for PHS and DH could be added in the analysis. 4. Page 6, line 42-43: Did all 1419 mothers deliver in only one DH out of 36? If it is true, then it is not a representative sample. Of the 36 DH, how many are included in the household survey? Clarify this in the text. 5. The results on availability of services are quite important. Among the reasons cited, unavailability of equipment is one of the reasons (page 7, line 25). Given the data, it is possible to analyze the correlation between the “services available/provided” and the equipment or structural capacity index. Authors could create an index for the services available and do a scatter plot. Service delivery are likely to be correlated with the staffing index and equipment index. 6. Kindly report the range (min and max) for infrastructure index and the equipment index on page 10-11. 7. The reported quality of care is better in PHC than the DH. What could be the possible reason for this? The structural capacity for DH is greater than PHC but the reported quality of care is better in PHC compared to DH. 8. The reported quality of care also suffers from recall bias. Mothers who gave birth 18 months ago may not recall the information accurately. Did you analyze Table 2 by the time of contact with PHC/DH? For example, divide the sample into two groups: mothers who delivered in the last one year versus (2) mothers who delivered 12-23 months before the survey. 9. Is it the “quality of care” or “perceived quality of care”? There is a difference between the two and authors should clarify whether they mean objective quality of care or perceived quality of care? 10. On the last page, authors conclude “This study lays.....”. readers would benefit from inclusion of prior studies on the association between quality of care an health outcomes. It should be reported either in the introduction or in the conclusion.
--	--

VERSION 1 – AUTHOR RESPONSE

Editor Comments to Author:

1. Please revise the strengths and limitations section after the abstract. This section should include methodological strengths as well as limitations.

Thank you. We have added two key points to this section. It reads as:

- *This study is limited to assessing the structural capacity of the facilities to deliver quality care and the care as reported by the mothers. However, there could be multiple other components that influence quality of care (e.g. skills and competencies of health personnel delivering care) that were not explored in this study.*
- *The study covers all DHs and a large representative sample of PHCs in Bihar. To our knowledge, no study of this scale has previously been conducted on facility readiness for maternal and newborn care in Bihar.*

2. Please revise your patient and public involvement statement so that it is in line with the description in our Instructions for Authors (<http://bmjopen.bmj.com/pages/authors/>). The statement should outline patient and public involvement in the planning and design of the study rather than outlining recruitment into the study. Please see our recent blog for further information regarding PPI <http://blogs.bmj.com/bmjopen/2018/03/23/new-requirements-for-patient-and-public-involvement-statements-in-bmj-open/>

There were no patients involved in the planning and design of the study. We have removed the previous statement and now it reads as

Patients were not involved in the study.

Reviewer(s)' Comments to Author:

Reviewer: 1

Reviewer Name: Deogratius Bintabara (MD)

Institution and Country: Department of Public Health, College of Health Sciences, The University of Dodoma, Tanzania

3. Please state any competing interests or state 'None declared': None declared

We have added 'None declared' on page 15.

Please leave your comments for the authors below

Thank you for the opportunity to review this informative paper. This is an important issue, and I congratulate the authors for bringing further attention to it. However, I have some slight amendments authors may wish to consider to improve the quality of the paper.

Thank you for reviewing this paper and giving us an opportunity to improve the quality of the paper. We have attempted to address the comments to the best of our ability.

General comments

Despite the fact that authors made their effort in their good work, the manuscript has got some weaknesses which need extensive revision. In Introduction, authors should try to state what the research gap(s) is. In Methods, the section missing a lot of important information that could help readers to follow the manuscript. Also, the sampling procedure is not clear and missing clear information about how the outcome variable has been derived. Missing of clear research questions lead authors to presents a lot of unnecessary findings in the results section. Furthermore, there were a lot of redundancies in methods, results, and discussion sections. Finally, the conclusion is weak. If the authors revised these areas probably they would have a very nice article.

Below are my comments

--Abstract:

4. Conclusion: Line 4-5, "....., this study revealed that improvement in service readiness is essential...." About this sentence, I think the study has to reveal the weakness or strength of what the authors assessed regarding their research questions and not what the others think needs to be improved.

We have changed the sentence to

Examining all DHs and a representative sample of PHCs in Bihar, this study revealed the gaps in structural components that need to be filled to provide quality care to mothers and newborns.

5. The same section, the second sentence "Access to quality care" despite the sentence looks ectopic but also difficult to understand what authors want to communicate. For example, reducing mortality of whom? I suggest the authors conclude based on their findings and provide recommendations on what has to be done based on their findings rather than providing a general sentence like this.

Thank you for your comment. The text now reads:

*Access to quality care is essential if progress in reducing **maternal and neonatal** mortality is to be achieved in this high-burden state.*

--Introduction:

6. The first paragraph, Line 3, the authors forgot to report MMR and MMR appropriately. The words "live births" are missing.

We have changed the text and it now reads:

Between 1990 and 2015, maternal mortality reduced from 556 to 174 per 100,000 live births and neonatal mortality reduced from 54 to 29 per 1,000 live births.

7. Many of the sentences in this section do not have the supporting references, therefore it is difficult to understand whether it is the known facts or just authors arguments.

Thank you for pointing this out. The reference for neonatal mortality was missing. We have added the following two references.

Between 1990 and 2015, maternal mortality reduced from 556 to 174 per 100,000 live births and neonatal mortality reduced from 54 to 29 per 1,000 live births.[1–3]

- *Lim, S. S., Dandona, L., Hoisington, J. A., James, S. L., Hogan, M. C., & Gakidou, E. (2010). India's Janani Suraksha Yojana, a conditional cash transfer programme to increase births in health facilities: an impact evaluation. The Lancet, 375(9730), 2009-2023.*
- *GBD 2015 Child Mortality Collaborators. Global, regional, national, and selected subnational levels of stillbirths, neonatal, infant, and under-5 mortality, 1980-2015: a systematic analysis for the global burden of disease study 2015. Lancet. 2016;388:1725–74.*

8. Paragraph 3, the authors stated that "gaps in readiness of facilities to provide quality care have been highlighted as an important barrier...." If this is known already what was the rationale of the current study? This can be achieved by clearly stating what is known so far about readiness to provide maternal and newborn care and what is not known that your research is going to cover.

The studies emphasize that programmes like JSY have not had the intended effect on outcomes like MMR due to poor quality of care and that facility readiness is an important area for exploration; but these studies do not describe and quantify the gaps. We have re-worded the text as:

Despite the success of JSY in increasing institutional deliveries, provision of quality care has been highlighted as an important barrier for the programme to have the intended effect on health outcomes.[11,12] Addressing the gaps in facility readiness has been considered as an important factor in improving quality of care.[13] Poor availability of clinical services due to lack of infrastructure has been recognized as one of the most common bottleneck to providing essential maternal and newborn services in India.[14] Rammohan et al. 2013 report lack of transport facilities for pregnant women as one of the major bottlenecks to access emergency obstetric care in India.[15] Capacity in terms of equipment and staff availability needs to be built to detect and manage obstetric emergencies.[16]

It is crucial to ensure that facilities are adequately resourced and equipped to deliver essential maternal and newborn care.[14,17–19] The role of quality factors such as infrastructure, equipment, supplies and staffing is acknowledged, but little research has been done to quantify and describe these gaps in detail. This is needed if interventions to strengthen quality are to be appropriately designed and targeted to be effective.[20]

--Methods:

Study population and sampling

9. Page 3, second paragraph; the sentence “Household survey data were collected from” Did you use the different questionnaires for the different age groups you identified? If no, why is it necessary to indicate age groups here? If you say “less than 24 months” will it show different meaning from what you have explained by including age group categories?

Yes, we used different questionnaires for different age groups and we used information that was collected only for mothers with children under 0 to 2 months. We have changed the text to:

*Household survey data were collected using **five different questionnaires** for mothers who had a child belonging to the following five age groups: i) 0-2; ii) 3-5; iii) 6-8; iv) 9-11; and v) 12-23 months old, respectively.*

We have added another sentence in the next paragraph as:

To reduce the recall bias, data on quality care were collected only from mothers with children aged between 0-2 months.

10. Page 3, Line 36-7, if the blocks were sampled proportionally according to the total number of blocks per district, how did it come resulting in 5-6 blocks sampled per district? Because this seems like an equal number of blocks per each district. The sentence is confusing.

Thank you for pointing this out; your comment is well received. The number of blocks vary widely per district and the blocks were allocated proportionally according to the blocks per district. The average blocks per district sampled were 5.9. We agree that this was worded in a confusing manner and thus we have updated the text:

The number of blocks vary widely per district. Hence, blocks were sampled proportionally according to the total number of blocks per district. The selected sample had blocks ranging from 1 to 9 per district with a median of 6 blocks.

11. Page 3, Line 5-7, in the previous paragraph you stated that the blocks were sampled proportionally according to the total number of blocks per district, here again, you stated that the number of AWC from each block was determined using proportional assumption. So how many AWC did you select per each block? Did you select an equal number of AWC per each block? In total how many AWC was selected?

Household survey data followed multi-staged sampling. All districts and blocks were included in this data. The number of Anganwadi Centers (AWC, village level institutions providing basic health care services) sampled from each block was determined using proportional allocation, however if this resulted in a sample of less than 19 AWCs, then 19 AWCs were sampled in order to meet a minimum sample threshold per block. The sampled AWCs were selected randomly within each block. 5 households per AWC were selected, with one each from mother of following five age groups- (a) 0-2 (b) 3-5 (c) 6-8 (d) 9-11 (e) 12-23. In total, 15667 AWCs were selected ranging from 19 to 123 per block.

The above information has been included in the manuscript.

12. The second paragraph in the same section, data collectors visited every 10th household looking for eligible mothers. Why after every 10th household? Why not 5th, 8th or 15th? It is important to explain the justification for doing that.

The data collection team conducted a pilot to check the distribution of households across the selected AWCs to understand if the geographical spread of the households provided any variation in the household background characteristics. During the pilot, no significant variation was observed between 5th, 10th or 15th household selection and to keep it operationally easy, every 5th household was chosen.

Following text has been added to clarify:

Starting with the index household, data collectors visited every fifth household looking for eligible mothers. This approach aimed to obtain a wide distribution of households (minimizing the effect of clustering), while remaining feasible and practical for data collection purposes. The pilot phase of the study did not observe any significant differences in household characteristics when alternative sample intervals of 10th, 15th, and 20th households were selected.

13. Similarly, why did you decide to include only those women with children between 0 – 2 months?

The study sought to understand the experience of quality care of women during the delivery of their children. Women who had given birth in the previous 0-2 months were selected for inclusion for two reasons: 1) We sought to describe services provided recently as a proxy of the current quality of care available; 2) recall bias is known to increase the longer ago an event occurred (thus, non-inclusion of women who gave birth in these facilities >2 months ago).

Data analysis

14. Generally, this section is not appropriately written. Instead of writing how they analyzed their data, authors just explaining the measurement of variables.

The measurement of variables as collected/measured in raw data is described in the data collection section. The data analysis section focuses on all key aspects of data analysis, both generation of scores and association analyses. It is crucial for the interpretation of the presented findings that these scores and other aspects of data manipulation are clearly described in the data analysis section. This

is as per standard journal guidance on manuscript writing. We are happy to receive further guidance from the reviewer on how to make this section clearer.

15. The section is poorly written and make it very difficult to follow.

We have reviewed this section and made changes where possible to improve understanding of the approach taken. We are very happy to respond to other suggestions from the reviewer on how to improve the section.

16. In the second paragraph, the authors reported that readiness was defined as availability. However, readiness and availability are two separate components used in assessing the capacity of the facility to provide services. The authors need to provide evidence of defining readiness based on availability.

Thank you for your comment. We have used the definition of readiness based on Service availability and readiness assessment (SARA) by World Health Organisation(WHO) SARA defines general service readiness as “the availability of components required to provide services , such as basic amenities, basic equipment, standard precautions for infection prevention, diagnostic capacity and essential medicines”.¹ We have added the reference to the manuscript.

17. Also, there is a lot of redundancies, for example, the last sentence of the first paragraph and the first sentence of the second paragraph explain the same thing.

The last sentence of the first paragraph reads:

“The current status of the facilities was assessed on three broad parameters, namely, the structural capacity, staffing, and the quality of care provided at the facilities.”

And the first sentence of the second paragraph reads:

“The structural capacity of the facilities was assessed by computing readiness scores of 0-1 for infrastructure, equipment and supplies using a set of questions included in our facility assessment tool.”

We are unsure where redundancy has been observed and would be grateful for further guidance from the reviewer on where this has occurred.

18. With the current information, it is not clear what is the outcome variable of this study. To me, it seems the outcome variable is "readiness of the facility to provide quality maternal and newborn care." However, there is no information in this paper that explains the overall readiness of each of the three predetermined indices or the global score. This is what I mean, based on those three indices how did you consider the facility is ready to provide the maternal and newborns care? This should be explained clearly.

Thank you for your comment. Assessing the readiness of the facility to provide quality maternal and newborn care is the primary outcome of this study. The overall structural capacity score at the PHCs and DHs and the staffing score provide the extent to which the facilities are ready to provide quality services. The study focuses on drugs, equipment, staff which are relevant to provide maternal and

¹ https://www.who.int/healthinfo/systems/SARA_Reference_Manual_Full.pdf

newborn care. The first paragraph on page 9 provides the readiness of each of the three components of structural capacity with the text as below:

Overall, the average structural capacity across PHCs was 60% (range: 35-83%) and 66% (range: 51-82%) across DHs. DHs were slightly better equipped on average in terms of availability of infrastructure, equipment and drug supplies (78%, 70%, 53%, respectively) compared with PHCs (63%, 65%, and 50%, respectively). Results varied greatly between facilities, particularly among PHCs.

If there is any additional information that should be further added, then we request the reviewer for further guidance.

19. The authors are required to provide information on how they merged the two data sets “facility dataset” and “household dataset.” Also, to state what was the unit of analysis in this study?

We have added this on page 6 under Quality of care sub-section. The text reads as

Household survey data was merged with facility data by matching the names of facilities where mothers delivered with the facility names collected during facility assessment survey. A quality of care index for each PHC and DH was generated by taking the average score of the 11 questions for all those household survey participants who delivered within the facility. All data were assessed at the facility level.

20. Based on your sampling technique, it seems you did a multistage (multilevel) sampling. During analysis did you consider to correct for “standard errors” as results of this kind of sampling you employed?

Thank you for your comment. We had run the analysis using multi-level mixed-methods regression analysis to adjust for clustering at the district level. We found no difference in the results that would lead to an alternative interpretation of our findings. In order to ensure easy interpretation of the findings by a wide audience (as anticipated for BMJ Open readership), we have presented unadjusted estimates.

Ethics and permission

21. The information regarding “informed consent and how was it performed” is missing.

Thank you. We have added details as follows:

Ethical approval was granted by the Indian Institutional Review Board. At each facility, the purpose of the study was explained and informed consent was obtained from the MOIC and the Hospital Manager or equivalent authority in charge in the PHCs and DHs, respectively. For the household survey, ethics approval was obtained from Ashirwad Ethics Committee, Ashirwad Hospital and Research Center, Ulhasnagar, India and informed consent was taken from the mothers.

--Results:

22. Generally, missing clear research questions lead authors to presents a lot of unnecessary findings in the results section.

Thank you for your comment. We hope that the added text below clarifies the objective and provides clear context to the relevance of the findings reported.

The first sentence of the last para of the intro section says:

This study aims to: (a) identify and quantify structural and staffing gaps in the public health facilities, specifically PHCs and DHs in Bihar, that need to be addressed to deliver quality maternal and newborn services; and (b) understand the relationship between structural and process quality metrics for maternal and newborn health services.

23. Page 6, Paragraph starting with "Facility survey data were" Please try to avoid redundancies some of the information here have also provided in the Methods section.

Thank you for this observation. We have removed the aforementioned redundant sentence to improve clarity and brevity.

24. In the sections of Facility characteristics and availability of services, there is a lot of unnecessary findings presented in the text.

We thank the reviewer for the comment but feel that this level of detail is required for readers who might be less familiar with health facility settings in India. In addition, in order for improvements to be made, we feel that it is important that adequate actionable detail is provided to the readers who may include those working within or responsible for healthcare provision in Bihar. The section provides data on the characteristics of facilities in Bihar and if they meet some of the basic recommendations by Indian Public Health Standard.

25. Page 9, the structural capacity has three components infrastructure, equipment, and drugs, but here you provided the average structural capacity in which you did not explain how you calculated it neither in the main document nor in the appendix.

Thank you. We have added the explanation in the methods sections and the text reads as:

The mean across the three components of infrastructure, equipment and supplies was computed to generate a score for structural capacity ranging from 0 to 1 per facility. The mean across facilities was computed to calculate an overall score for structural capacity.

26. Page 9 Paragraph 2 and 3, and Page 10, Paragraph 1, 2, and 3, there is a lot of repetition of information provided in the methods section.

We have reviewed these sections and minimised inclusion of any methods text were possible. However, in some cases, reference to methods has been included to avoid confusion in interpretation of the results. We appreciate that there are many components in this study, with a lot of data from many variables. In light of this, we have made every effort to minimise confusion of the reader. It has, therefore, been necessary, in places, to point the reader towards the section of the data collection from which the results are coming. E.g. "Of the 30 services assessed in 36 DHs and 189 PHCs,..." If the reviewer and editors feel that further editing is required, we are very happy to receive further comments with suggestions and to make additional modifications.

Furthermore, redundancies of results of table 1 and texts.

The text corresponding to table 1 is providing more details on what the IPHS recommendations are and also giving information on contractual workers for some cadres. We have removed the sentence “Almost a third (30%) of the PHCs did not have a pharmacist” since this is evident in the table.

27. What is the relevance of assessing the relationship between structural capacity and staffing, structural capacity and quality of care, and staffing and quality of care? For example, there was a clear relationship, then, what did you want to tell the readers.

The literature on quality of care has often theorized about the interrelationship between various components of quality². For example, low levels of staffing lead to low quality of care. Yet, little data on this issue have actually been reported. Furthermore, the analysis provides insights into clustering of quality indicators -e.g if some facilities are doing poorly on all metrics and some are doing well, then one would expect a high degree of correlation between these factors. If different facilities experience different challenges, then there will not be any correlation. The quality improvement interventions needed differ in these differing scenarios.

We have highlighted this in the conclusion with the following text:

Lack of correlation between structural capacity and staffing, and structural capacity and quality of care suggests presence of heterogeneity in the strengths and weaknesses across the facilities.

--Discussion:

28. Generally, the discussion is weak and the authors failed to discuss in detail the important findings from this study instead they repeated to explain them as what they did in the results section.

--Conclusion:

29. The conclusion needs to be revised to address the gap or the strength observed instead of recommending by using the general information not based on the results they obtained.

We have sought to minimise repetition in the discussion and conclusion section. Having provided a summary of key findings in the opening paragraph of the discussion, the conclusion focuses on interpretation and relevance of the findings. This style is the most common structure applied to a discussion section for medical academic journals, including BMJ Open.

Reviewer: 2

Reviewer Name: Santosh Kumar

Institution and Country: Sam Houston State University, Texas, USA.

Please state any competing interests or state 'None declared': I declare that i have no competing interest.

Please leave your comments for the authors below

BMJ Open

Manuscript ID: bmjopen-2018-028370

Summary:

The authors report a descriptive analysis of readiness of primary health centers (PHC) and district hospitals (DH) to provide maternal and newborn care in Bihar, the third most populous state with poor health outcomes in India. Based on facility and household survey, the authors construct structural capacity, staffing index, and quality of care index. Surprisingly, the authors did not find any clear correlation between facility structural capacity index, staffing index, and quality of care.

The manuscript is well written and reports the survey findings on readiness of health clinics that are

² Kruk, M. E., Gage, A. D., Arsenaault, C., Jordan, K., Leslie, H. H., Roder-DeWan, S., ... & English, M. (2018). High-quality health systems in the Sustainable Development Goals era: time for a revolution. *The Lancet Global Health*, 6(11), e1196-e1252.

essential for improving maternal and newborn care.

Comments:

Thank you for reviewing our manuscript. The comments provided by the reviewer are extremely insightful and have helped us to improve the quality of this manuscript. We have attempted to address the comments to the best of our ability.

30. Introduction needs to be strengthened by including some empirical evidence on the facility readiness and facility volume of services. Since the results show correlation between structural capacity & staffing / quality of care etc, authors should report relevant literature on these associations. As of now, it lacks literature on these correlations.

Thank you for your comment. We have added more literature in the introduction section that highlights the relevance of structural capacity and staffing with quality of care. The text now reads as:

Addressing the gaps in facility readiness has been considered as an important factor in improving quality of care.[13] Poor availability of clinical services due to lack of infrastructure has been recognized as one of the most common bottleneck to providing essential maternal and new born services in India.[14] Rammohan et al. 2013 report lack of transport facilities for pregnant women as one of the major bottlenecks to access emergency obstetric care in India.[15] Capacity in terms of equipment and staff availability needs to be built to detect and manage obstetric emergencies.[16]

It is crucial to ensure that facilities are adequately resourced and equipped to deliver essential maternal and newborn care.[14,17–19] The role of quality factors such as infrastructure, equipment, supplies and staffing is acknowledged, but little research has been done to quantify and describe these gaps in detail. This is needed if interventions to strengthen quality are to be appropriately designed and targeted to be effective.[20]

31. How the average structural capacity is across PHC and DH constructed? Is it the simple average of the three scores, namely, drugs, equipment, infrastructure. Could it be greater than one? What is the range of the structural capacity index? Please clarify this on page 9, line 5-6.

Thank you, we have added the details on page 5 under data analysis. The text now reads as:

The mean across the three components of infrastructure, equipment and supplies was computed to generate a score for structural capacity ranging from 0 to 1 per facility. The mean across facilities was computed to get an overall score for structural capacity.

32. The facility survey data are from 2016. A similar facility survey was done in District Level Household Survey (round 3) in 2007-08. It would be interesting to see the trend in facility readiness over time. A comparative analysis of DLHS 3 and BTSP survey for PHS and DH could be added in the analysis.

We thank the reviewer for this excellent suggestion. We feel that this question deserves attention in its own right as a separate analysis. We will certainly explore this question for a future manuscript.

33. Page 6, line 42-43: Did all 1419 mothers deliver in only one DH out of 36? If it is true, then it is not a representative sample. Of the 36 DH, how many are included in the household survey? Clarify this in the text.

All 1419 mothers delivered in a DH, with a spread across all 36 DHs. The text has been edited to clarify this.

Household survey data were available from 671 mothers who delivered in 107 of the 190 study PHCs and 1419 mothers who delivered across all 36 study DHs.” The authors meant that these 1419 mothers delivered in any of these 36 DHs.

34. The results on availability of services are quite important. Among the reasons cited, unavailability of equipment is one of the reasons (page 7, line 25). Given the data, it is possible to analyze the correlation between the “services available/provided” and the equipment or structural capacity index. Authors could create an index for the services available and do a scatter plot. Service delivery are likely to be correlated with the staffing index and equipment index.

Thank you. We have added these results

The methods section has been updated with the following text

The relationship between availability of services (that were unavailable in at least 10% of the PHCs and DHs) and structural capacity and staffing index was explored by assessing the pairwise correlation coefficients between the indices at the facility level.

The results section has been updated with:

Relationship between service availability indexes

In PHCs, all three components of structural capacity index, including infrastructure, equipment and supplies, had significantly positive correlation with the availability of the 12 services at 5% level of significance. For DHs, availability of seven services that were unavailable in at least 10% DHs, had positive correlation with equipment, supplies and staffing index at 5% level of significance.

35. Kindly report the range (min and max) for infrastructure index and the equipment index on page 10-11.

Thank you for your comment. We have added the range under Infrastructure sub-heading on page 9 as follows:

Infrastructure score at the DHs and PHCs varied with a range of 60-92% and 32-90%, respectively.

We have added under Equipment and Supplies sub-heading on page 10:

Equipment score at DHs ranged between 54% and 92%; the corresponding range at PHCs was 40-90%. Drug score varied with a range of 25-100% across DHs and 14-82% across PHCs.

36. The reported quality of care is better in PHC than the DH. What could be the possible reason for this? The structural capacity for DH is greater than PHC but the reported quality of care is better in PHC compared to DH.

There could be other factors contributing to quality of care at the facilities such as the case load, availability of trained and qualified staff etc. Usually the DHs have a very high case load. The ratio of case load to staff available could be one factor contributing to the level of quality care provided. Unfortunately, we did not have granulate data on case load in order to explore this question in this analysis.

We have added the text to explain this in the discussion section. The text reads as:

The results revealed that DHs, on average, were better in terms of staffing and structural capacity in comparison with PHCs. However, the reported quality of care was better in PHCs than the DHs. DHs, being the referral points for PHCs, often need to address complicated cases and are therefore recommended to have higher staffing and structural capacity in comparison to PHCs. However, the quality of care provided at DHs and PHCs would also depend on other factors including the case load and type of cases.

37. The reported quality of care also suffers from recall bias. Mothers who gave birth 18 months ago may not recall the information accurately. Did you analyze Table 2 by the time of contact with PHC/DH? For example, divide the sample into two groups: mothers who delivered in the last one year versus (2) mothers who delivered 12-23 months before the survey.

The quality of care data in this study was only collected from mothers with children between 0-2 months. The 2 month restriction was selected in order to minimize the effect of recall bias, as mentioned by the reviewer.

38. Is it the “quality of care” or “perceived quality of care”? There is a difference between the two and authors should clarify whether they mean objective quality of care or perceived quality of care?

This is reported quality of care based on women’s reported experience. This is indeed a perception of quality of care. However, it is also worth noting that the reported experience of patients receiving care is in itself an important metric of quality³.

We clarify this in the text and it reads as:

We also conducted analyses of household survey data to explore the quality of care at facilities as reported by women who both participated in the household survey and delivered at study facilities.

39. On the last page, authors conclude “This study lays.....”. readers would benefit from inclusion of prior studies on the association between quality of care and health outcomes. It should be reported either in the introduction or in the conclusion.

The literature has been added in the introduction section as follows:

Addressing the gaps in facility readiness has been considered as an important factor in improving quality of care.[13] Poor availability of clinical services due to lack of infrastructure has been recognized as one of the most common bottleneck to providing essential maternal and new born services in India.[14] Rammohan et al. 2013 report lack of transport facilities for pregnant women as one of the major bottlenecks to access emergency obstetric care in India.[15] Capacity in terms of equipment and staff availability needs to be built to detect and manage obstetric emergencies.[16]

It is crucial to ensure that facilities are adequately resourced and equipped to deliver essential maternal and newborn care.[14,17–19] The role of quality factors such as infrastructure, equipment, supplies and staffing is acknowledged, but little research has been done to quantify and describe these gaps in detail. This is needed if interventions to strengthen quality are to be appropriately designed and targeted to be effective.[20]

³ Sanghita Bhattacharyya, Aradhana Srivastava & Bilal Iqbal Avan (2013) Delivery should happen soon and my pain will be reduced: understanding women's perception of good delivery care in India, *Global Health Action*, 6:1, 22635, DOI: 10.3402/gha.v6i0.22635

VERSION 2 – REVIEW

REVIEWER	Deogratus Bintabara The University of Dodoma, Tanzania
REVIEW RETURNED	04-May-2019
GENERAL COMMENTS	Authors tried to their best level to address all the comments